# Millennial scale sea surface temperatures of the western Arabian Sea between 37 - 67 ka BP

Jennifer Scott[1,2], Douglas Coenen[3,4,] Simon Jung[2]

[1] School of Engineering and Physical Science, Heriot Watt University, Edinburgh, EH14 4AS, United Kingdom
[2] School of Geosciences, University of Edinburgh, Edinburgh, EH9 3FE, United Kingdom
[3] Institute of Geosciences, Goethe University Frankfurt, Frankfurt am Main, Germany,
[4] Frankfurt Isotope and Element Research Center (FIERCE), Goethe University Frankfurt, Frankfurt am Main, Germany

*Correspondence to*: Jennifer Scott (js2092@hw.ac.uk) or Simon Jung (simon.jung@ed.ac.uk)

**Abstract**. The Asian monsoon system is a crucial part of the global climate system affecting a significant proportion of the world population. Understanding the controls for changes in the monsoon system is crucial for meaningful assessments of future climate change. The Arabian Sea is part of the wider Asian monsoon system and has been studied regarding controls of monsoon variability through time. In this study we present sea surface temperature data from 37 - 67 ka BP from sediment core NIOP 929 from the western Arabian Sea assessing the importance of
northern/southern hemispheric climate change driving monsoon circulation in the Arabian Sea. Earlier work implies a straightforward link between monsoon variation in the Arabian Sea and northern hemisphere millennial scale climate change during glacial periods, as depicted in Greenland ice cores. We present a new millennial-scale Mg/Ca based sea surface temperature reconstruction based on the planktic foraminifera species *G. bulloides* and *G. ruber*. We use these data to calculate seasonal sea surface temperatures. The SST data are variable with a maximum short-
term change of 8-9 °C. The variations in our SST records appear not related to change in either hemisphere in a straightforward fashion by not showing a phase-locked relation to millennial scale change in Greenland or Antarctic ice core records. We discuss these changes in the context of the Arabian Sea potentially being a "melting pot" with both the northern and the southern hemisphere exerting influence on a seasonal scale.

## 1.      Introduction

The climate and surface ocean conditions of the Arabian Sea are controlled by the seasonally reversing Asian monsoon winds (Brock et al., 1992; Prell and Curry, 1981; Rixen et al., 1996; Shetye et al., 1994). The change from the summer southwest monsoon (SW monsoon) to the winter northeast monsoon (NE monsoon) entails seasonally

and spatially varying patterns in sea surface temperature/circulation, nutrient distribution and biological productivity (Anand et al., 2008; Conan and Brummer, 2000; Curry et al., 1992; Ivanochko et al., 2005; Jung et al., 2002; Peeters

et al., 2002; Rostek et al., 1997; Saher et al., 2007b; Saher et al., 2007a; Saher et al., 2009; Schulz et al., 1998). The SW monsoon in particular is socio-economically significant because it drives rainfall on the Indian subcontinent, affecting the livelihood of billions of people through agriculture and water resources (Gadgil and Gadgil, 2006; Mishra et al., 2020; Webster et al., 1998).

The SW monsoon is caused by differential land-ocean heating and the resulting atmospheric pressure gradients

between the Indian subcontinent and the Indian Ocean. During summer, development of a low-pressure cell over the Tibetan Plateau causes the intertropical convergence zone (ITCZ) to shift northward, inducing south-westerly winds in the western Arabian Sea (Schott and Mccreary, 2001) (Fig. 1). These winds drive upwelling along the coast of Somalia and Oman through Ekman transport (Clemens et al., 1991; Wyrtki, 1973) (Fig. 1, 2a). Modern-day sea surface temperatures (SSTs) have been recorded as low as 16°C during the summer SW monsoon (Swallow and

Bruce, 1966), reflecting the cold, nutrient rich waters surfacing from a depth of around 200m (Peeters et al., 2002) (Fig. 2a). During winter the pressure gradient reverses and the ITCZ shifts southward, inducing northeasterly monsoon winds in the Arabian Sea (Fig. 1). No upwelling occurs off Somalia during the winter and the inter-monsoon periods (Fig. 2b), entailing generally higher SSTs; typically, in the modern-day between 25–28°C but can reach over 30°C (Swallow and Bruce, 1966).

The sedimentary record reflects the seasonally reversing upwelling signal and has been used to reconstruct monsoon variability on glacial-interglacial timescales, with a stronger SW monsoon being recorded as lower sea surface temperatures and increased productivity (Rostek et al., 1997). Productivity proxies used to investigate monsoon variability include abundance pattern in planktonic foraminifera (Anderson and Prell, 1993; Gupta et al., 2003; Kroon, 1991), $\delta^{18}O$ (Prell et al., 1992), total organic carbon content (Caley et al., 2013; Ivanochko et al., 2005;

Schulz et al., 1998), $\delta^{15}N$ (Altabet et al., 1995; Ivanochko et al., 2005), Ba/Al ratios (Ivanochko et al., 2005), bromine records (Caley et al., 2013) and changes in coccolithophore assemblages (Beaufort et al., 1997). Several authors have studied the glacial-interglacial SST history of the western Arabian Sea, using alkenones (Rostek et al., 1997; Saher et al., 2009; Sonzogni et al., 1997) and Mg/Ca derived calcification temperatures of planktonic foraminifera (Anand et al., 2008; Ganssen et al., 2011; Saher et al., 2007b; Saher et al., 2007a; Saher et al., 2009). These studies

support the notion that summer monsoon induced upwelling may have been slightly weakened during the last glacial, but principally persisted throughout this period. Compared to glacial-interglacial SST reconstructions there is a lack of high-resolution, centennial to millennial scale records from the region, with the only available data sets covering the last deglaciation (Anand et al., 2008; Saher et al., 2007b; Saher et al., 2007a) and the previous interglacial (Saher

et al., 2009). In this study we will address this shortfall by presenting a millennial scale Mg/Ca derived SST record for *Globigerinoides ruber* and *Globigerina bulloides* from sediment core NIOP 929 spanning 37 to 67 ka BP. A novel proxy is employed to reconstruct the winter and summer monsoon sea surface temperatures using these foraminiferal Mg/Ca based temperatures.

## 2.     Methods

### 2.1.     Core NIOP 929

Sediment core NIOP 929 was recovered during the 1992 - 1993 Netherlands Indian Ocean Programme (NIOP) (Van Hinte et al., 1995). The 16.15m long core was taken from the western Arabian Sea (Fig. 1) (13°42 21 N, 53°14 76 E) at 2490m water depth, with the sampling location being within the modern day summer monsoon induced upwelling area off the coast of Somalia (Rostek et al., 1997; Saher et al., 2007a).

### 2.2.     $\delta^{18}O$ record

Sampling of core NIOP 929 involved continuous slicing of the top 210 cm of the sediment sequence in 0.5cm steps. Below this depth sampling continued at 1cm steps. The stable oxygen isotope data from the top 210 cm used in this study have been published previously (Saher et al., 2007b; Saher et al., 2007a). The remainder has been measured at the University of Edinburgh and are reported on here. Samples were freeze dried and wet-washed over a 63µm sieve before 25-30 specimens from the 250 -355 µm size fraction of both *G. ruber* and *G. bulloides* were picked for stable isotope analysis. In rare cases where samples did not contain sufficient specimen less than 25 specimen were analysed. Sample preparation for stable isotope analysis followed standard procedures. The stable isotope analyses for the top 210 cm of core NIOP 929 were carried out at the Vrije Universiteit Amsterdam using a Finnigan MAT252 equipped with a Kiel device. Deeper sections of the core used in this study were measured at the School of Geosciences at the University of Edinburgh with a Thermo Electron Delta+ Advantage stable isotope mass spectrometer coupled to a Kiel Carbonate III preparation device. The reproducibility of the Amsterdam oxygen isotope data is 0.08‰ and that of the Edinburgh data is 0.1‰.

## 2.3.    NIOP 929 age model

The age model for core NIOP 929 is based on a stepwise approach (compare Fig. 3). In step 1, the age model for the upper 210 cm of core NIOP 929 was established based on seven 14C ages from (Saher et al., 2007b). The 14C ages were recalibrated to calendar ages using the 14C age calibration program Calib 8.2 (Stuiver and Reimer, 1993), using a ΔR value of 45 (±67) (Southon et al., 2002) (Tab. 1, Fig. 3). In steps 2 and 3, the age model of samples deeper than 210 cm was developed based on oxygen isotope stratigraphy. The NIOP 929 *G. ruber* $\delta^{18}$O record was initially tuned to the LR04 $\delta^{18}$O global stack of (Lisiecki and Raymo, 2005)(step 2), as the population of *G. ruber* reflects annual average conditions of the study site (Conan and Brummer, 2000). The LR04 age control points are shown in Tab. 2. In the final (third) step, this initial age model was refined by tuning to the Greenland ice core NGRIP $\delta^{18}$O record (Andersen et al., 2006; Rasmussen et al., 2006; Svensson et al., 2006; Vinther et al., 2006; Wolff et al., 2010)(the combined set of tuning based age control points is shown in Tab. 3). This approach is supported by earlier findings implying that monsoonal change at the millennial scale in the Arabian Sea occurred in phase with millennial scale change in the North Atlantic region(Altabet et al., 1995; Ivanochko et al., 2005; Jung et al., 2002; Schulz et al., 1998) (Fig. 3). Both the NGRIP $\delta^{18}$O record and the initially age modelled NIOP 929 *G. ruber* $\delta^{18}$O record were filtered to a 700-yr running mean before tuning to better constrain the expression of millennial scale climate variability in the records (Zeeden et al., 2020) (Fig. 3).

The uncertainties in our age model depend on the nature of the age control points. AMS14C based age estimates entail 1 sigma uncertainty ranges between ~200 and ~350 years (Tab. 1). The 2 sigma age envelopes range from ~410 to ~710 years (Tab. 1). The maximum layer counting error for the NGRIP record varies between 7% during the cold stadial events and 4% during the warm interstadial events (Andersen et al., 2006). The uncertainty in the LR04 record varies throughout the 5 Ma record. For the youngest section, i.e. 1-0 Ma it is around 4 ka (Lisiecki and Raymo, 2005). Whilst these uncertainties limit the accuracy of absolute dating of specific events beyond the capabilities of the individual dating method, the comparability of the sequence of events in our study is also depended on the quality of the visual alignment of events within the temporal framework determined by the reference record. It is difficult to quantify this uncertainty, but our best conservative guess that it is in the region of 2-3 ka. Whilst this is potentially a reason for concern when comparing short-term events, in the absence of absolute dating methods covering the entirety of the record presented in this study, it is the best that can be done.

## 2.4.    Mg/Ca paleothermometry

Sediment samples for Mg/Ca paleothermometry were obtained from core NIOP 929 from 387.5 - 643.5 cm depth (Van Hinte et al., 1995). Availability of samples did not allow for continuous sampling at 1cm spacing. No samples were available from 644.5 - 655 cm depth due to sample loss in core recovery (Van Hinte et al., 1995). Twenty-five tests of both *Globigerinoides ruber* (white) and *Globigerinoides bulloides* from the 250 - 355 µm size fraction were picked for Mg/Ca analysis. Samples were prepared using the method described by (Barker et al., 2003), with an

additional MilliQ rinse and ultrasonification prior to clay removal instead of crushing samples due to high levels of sample loss. Samples were analysed at the School of GeoSciences at University of Edinburgh on a Varian ICP-OES Vista Pro (De Villiers et al., 2002). Instrumental precision was ± 1% based on replicate measurements of the ECRM 752-1 carbonate standard. The measured Mg/Ca concentrations were then converted to temperatures using Eq. (1) for *G. ruber* (Anand et al., 2003) and Eq. 2 for *G. bulloides* (Mashiotta et al., 1999):

$$Mg/Ca = 0.34\ e^{0.82T} \tag{1}$$

$$Mg/Ca = 0.47\ e^{0.107T} \tag{2}$$

where Mg/Ca is the measured Mg/Ca ratio of the sample and T the reconstructed temperature recorded in the foraminiferal calcite. The Mg/Ca temperature estimate of (Mashiotta et al., 1999) was used instead of (Vázquez Riveiros et al., 2016) as the temperature range of 10 - 25°C better suits the expected temperature range of the

environment (Vázquez Riveiros et al., 2016).

## 2.5.    Winter and summer temperature estimates

Previous work by (Saher et al., 2007a) used the δ[18]O values for both *G. ruber* and *G. bulloides* for core NIOP 929 in conjunction with Mg/Ca temperatures from *G. ruber* to reconstruct summer and winter monsoon temperatures for the past 20 ka BP. Part of their study involved analysing seasonal foraminiferal abundance data from nearby sediment

traps (Conan and Brummer, 2000; Curry et al., 1992; Nair et al., 1989) in order to define the *G. ruber* and *G. bulloides* flux throughout the year. A zero flux of both *G. ruber* and *G. bulloides* was assumed for the inter-monsoon periods, based on the sediment trap data. With r and b denoting the fractions of *G. ruber* and *G. bulloides* which calcify during the summer monsoon season, their analysis yielded results of 0.54 ± 0.08 for r and 0.91 ± 0.06 for b (Saher et al., 2007a).

As this study reconstructs both *G. ruber* and *G. bulloides* Mg/Ca temperatures, we can apply these seasonal abundance values to reconstruct winter and summer monsoon temperatures. As illustrated by (Saher et al., 2007a),

a population of *G. ruber* will record a temperature which can be attributed as 0.5 summer temperature and 0.5 winter temperature (Fig. 2) (Conan and Brummer, 2000). *G. bulloides* is predominantly present during the summer monsoon period, however a small portion is present within the winter monsoon period meaning the temperature signal recorded in a population can be inferred as 0.9 summer temperature and 0.1 winter temperature (Fig. 2) (Conan and Brummer, 2000). Therefore, winter and summer monsoon temperatures can be calculated from G. *ruber* and *G. bulloides* Mg/Ca temperatures as shown in Figure 2.

## 2.6.    Uncertainties in our SST data

The standard error for Mg/Ca based SST estimates using planktic foraminifera varies among different calibration studies, i.e. between±0.7°C (at the temperatures in our study; Vázquez Riveiros et al., 2016), ±0.8°C (Mashiotta et al., 1999) and ±1.2-1.4°C (established in different oceans and for different planktonic species with a similar habitat; Dekens et al. 2002). In order to visualize the uncertainty related to our SST data, Fig. 4 indicates the error envelope for the *G. ruber* and *G. bulloides* based SST records. The error envelopes reflect an uncertainty of ±0.8°C (Mashiotta et al., 1999; used in this work). We have applied the same uncertainty to the derived winter and summer temperatures. Calculating the latter two temperatures relies on the abundance of *G. ruber* and *G. bulloides* and their seasonal preferences when forming throughout the year. In the absence of data constraining the seasonal distribution pattern of either species through time, we assumed the ratio has been stable (as described above). Given the range of SST changes in our records (e.g. Fig. 4), whilst there are smaller variations that fall within the error envelopes, the larger excursions, with SST swings beyond the error envelopes seem robust. In this work we will focus on these large SST changes.

## 3.    Results

### 3.1.    δ¹⁸O records

The $\delta^{18}O$ records of *G. ruber* and *G. bulloides* from core NIOP 929 cover the time interval from 1.3 ka BP to 126 ka BP on a centennial scale (Fig. 3). The $\delta^{18}O$ *G. ruber* record has a range of values from 0.4‰ at 21.4 ka BP to -2.3‰ at 1.8 ka BP. The $\delta^{18}O$ *G. bulloides* record has a range of values from 0.3‰ at 22.7 ka BP to -2.1‰ at 8.5 ka BP. Within the time window between 35 and 70 ka BP (Fig.4) both stable oxygen isotope records are dominated by

millennial to sub-centennial oscillations. In the stable isotope record of *G. ruber* this variability is superimposed on weakly defined long-term minima centered at ~60 ka BP, ~52-53 ka BP, ~45-46 ka BP, with maxima being recorded in between. Weakly defined long-term minima in the stable oxygen isotope record of *G. bulloides* are centered at 165  ~57-58 ka BP, ~52 ka BP, ~44-46 ka BP and 38 ka BP, alternating with maxima occurring in between.

### 3.2.    Mg/Ca temperature reconstructions

The Mg/Ca based temperature records of *G. ruber* and *G. bulloides* from core NIOP 929 cover the time interval from 37.4 ka BP to 67.7 ka BP on a sub millennial scale (Fig. 4-6). The *G. ruber* Mg/Ca temperature record has a range of 9. 2°C from a low of 21.2 °C at 49.0 ka BP, to a high of 30.4°C at 39.2 ka BP, with a mean temperature of 23.7°C 170  (Fig. 4, 5, 6). The average temporal resolution for the *G. ruber* record over the reconstructed period is 203 yrs. The *G. bulliodes* Mg/Ca temperature record has a similarly large range in temperatures of 9.9°C, with the lowest value of 16.0°C at 62.5 ka BP, the highest value of 26.8°C at 64.8 ka BP and a mean temperature of 21.2°C (Fig. 4, 5, 6). The average temporal resolution for the *G. bulliodes* record is 216 yrs.

In Figure 4, we use 700 year box car filtered times series to focus on the most robust signal in our SST records. Both, 175  the *G. ruber* and the *G. bulloides* based SST records display several maximum and minimum spikes at a millennial scale. Prominent maximum spikes in the *G. ruber* SST records are centred at ~61 ka BP, 57-54 ka BP (two sub peaks), ~51 ka BP, ~42 ka BP, ~40.5 ka BP and ~39.5 ka BP, with the highest temperatures of >27°C being linked with the youngest maxima. Pronounced minima in the *G. ruber* SST record are centred at ~63 ka BP, ~57-57.5 ka BP, ~52 ka BP, ~49 ka BP (broad maxima), ~46 ka BP, ~44 ka BP, ~41.5 ka BP, ~40 ka BP and ~37 ka BP.

180  Some of the maximum spikes in the *G. ruber* record are roughly matched in the *G. bulloides* SST record (Fig. 4), but there are differences too. Distinct maxima in the *G. bulloides* SST record are centred at ~67 ka BP, ~65 ka BP, ~61 ka BP (broad maxima), ~52 ka BP, ~49-46 ka BP (two sub peaks), ~41.5 ka BP, ~40.5 ka BP and ~38.5 ka BP. Interestingly, contrasting the *G. ruber* SST record, the highest values the *G. bulloides* SST record appear near the beginning and the end of the record. Minima in the latter occur at ~66 ka BP, ~63 ka BP, ~57-57.5 ka BP, ~53.5 ka 185  BP, ~49.5 ka BP, ~45 ka BP, ~42.5 ka BP and ~39.5 ka BP, with amplitudes diminishing in the in the upper part of the record.

In relation to orbitally driven climate change, whilst not being at the centre of this study, it is interesting to note that both records suggest some similarity with summer insolation at 30ºN, with the *G. bulloides* SST record showing a slightly stronger affinity.

## 3.3.    Winter and summer temperature reconstructions

The Mg/Ca based temperature records of *G. ruber* and *G. bulloides* were used to reconstruct winter and summer monsoon temperatures from 37.4 ka BP to 67.7 ka BP (Fig. 4, 7). Winter temperature is lowest at 17.3°C at 64.8 ka BP and highest at 34.7°C at 37.5 ka BP, with an overall range of 17.4°C and mean temperature of 26.7°C (Fig. 7). Summer temperature is lowest at 15°C at 62.5 ka BP and highest at 27.7°C at 37.9 ka BP, with a range of 12.4°C and a mean temperature of 20.4°C (Fig. 7).

The general shapes of the 700 year box car filtered, season specific winter and summer temperature records are distinctly different (Fig. 4), by being dominated by (mostly) negative and positive excursions, respectively. For the majority of the winter SST record, maximum values are around 28 - 29°C in Fig. 4, with a maximum centred at 39.5 ka BP being a possible exception. Prominent minima in winter temperatures occur around 65 ka BP, ~52-52.5 ka BP, ~41.5 ka BP (single data point) and 38 ka BP. These minima are superimposed on a trend of generally increasing winter temperatures in the early part of the record. Whilst close to the detection limit of the record, the long-term component of the winter temperature time series resembles summer insolation at 65degN (Fig. 4).

For the majority of the summer temperature time series, minimum values are around 18 °C or slightly below. Prominent SST maxima occur at ~67 ka BP, ~65 ka BP, ~61-62 ka BP (broad), ~52 ka BP, ~49-46 ka BP (two sub peaks), ~42 ka BP, and ~38 ka BP. These short-term maxima are superimposed on a long-term oscillation with generally higher summer temperatures occurring near the beginning and the end of the sequence.

The longer-term variability in winter and summer temperatures can be subdivided into four periods (Fig. 4). In period I from 37 - 42 ka BP summer monsoon temperatures are on average ~3.7°C lower than the winter monsoon temperatures compared to a larger temperature difference in the preceding period. The reduced temperature difference in period I stems from two prominent excursions at ~41.5 ka BP and ~38 ka BP (Fig. 4) in both, summer and winter temperatures, showing maximum and minimal values, respectively. During both events (one of which only defined by one data point), the normal temperature gradient (sensu Swallow and Bruce, 1966) reversed with summer temperatures exceeding winter values by 4-6 degree in Fig. 4.

Period II from 42 – ~51.7 ka BP is more consistent with the seasonal temperature gradient of modern monsoon dynamics showing mean summer monsoon temperatures 7.4°C lower than winter values. During this period there are no instances where summer temperatures are higher than winter temperatures, with winter temperatures between 4.2 – 11.9°C higher than summer temperatures. During period III from 51.7 – 64.4 ka BP summer monsoon temperatures were on average 7.5°C lower than winter temperatures, amid short term variability in particular in the unsmoothed record (Fig. 7) with summer temperatures being up to 4.7°C higher than winter values. In the smoothed

record in Fig. 4, this is reflected by short periods with summer and winter temperature differences being close to zero, most prominently depicted at ~52 ka BP. During Period IV from 64.4 – 67.7 ka BP mean summer temperatures are ~0.3°C higher than winter values. There is, however, a large variability displayed during this period. Similar to period I, there are two prominent temperature excursions in both, summer and winter temperatures, showing maximum and minimal values, respectively. Both events, centred at ~65 ka BP and ~66.5 – 67 ka BP, entail a reversal

of the normal temperature gradient with summer temperatures exceeding winter values by ~4.5°C and ~1°C, respectively.

## 4. Discussion

### 4.1. Robustness of Mg/Ca based SST estimates in the Arabian Sea

Here we present the most complete temperature history of core NIOP 929 to date (Fig. 4, 5). The rapid SST changes

reconstructed in this study are greater than those of any previous reconstruction for core NIOP 929 (Fig. 4, 6). The 9.2°C range of temperatures reported in the *G. ruber* record of this study is much greater than the 5.3°C range in temperatures in *G. ruber* across the last deglaciation (Saher et al., 2007b; Saher et al., 2007a) and 4.2°C range across the penultimate interglacial (Saher et al., 2009).

The high-resolution Mg/Ca temperature reconstructions of this study from 37.4 ka BP to 67.7 ka BP can be directly

compared to the alkenone based Uk'37 temperature record of (Rostek et al., 1997). Alkenone Uk'37 temperature estimates have been shown to reflect annual average SSTs (Bijma et al., 2001; Budziak et al., 2000; Müller et al., 1998; Sonzogni et al., 1997), and with roughly equal fluxes of *G. ruber* calcifying in the winter and summer in the Arabian Sea (Conan and Brummer, 2000; Curry et al., 1992), the *G. ruber* Mg/Ca SST record can also be considered an annual average SST record. Overall, there is reasonably good agreement between the two proxies, with the

alkenone Uk'37 record of (Rostek et al., 1997) tracking the mean value of the higher resolution sections of this study's *G. ruber* Mg/Ca record (Fig. 6). For the lower resolution period 43 - 52 ka BP the records do not agree as well, with the alkenone Uk'37 temperature reports values around 2°C higher than those of the *G. ruber* Mg/Ca record (Fig. 6).

Previous studies have noted the disparity in the values of alkenone Uk'37 and Mg/Ca based temperature records for

core NIOP 929, with (Saher et al., 2007b) finding the alkenone Uk'37 based temperature estimates of (Rostek et al., 1997) around 2°C higher than their Mg/Ca *G. ruber* record for the last deglaciation, and (Saher et al., 2009) finding

a near constant offset of 3.5°C between the alkenone Uk'37 record of (Sonzogni et al., 1997) and their *G. ruber* Mg/Ca record during MIS 5 – 6 (Fig. 6). Similar offsets have been documented in the North Atlantic Ocean for the past 28 ka BP (Elderfield and Ganssen, 2000) and the Equatorial Atlantic Ocean for the last 270 ka BP (Nürnberg et

al., 2000). The transfer function used to convert alkenone saturation to temperature and the equation to convert Mg/Ca ratio to temperatures can affect the offset between the two records (Saher et al., 2009). As the alkenone saturation values of (Rostek et al., 1997) are not available, changing the transfer function used to investigate the potential difference is not an option. Using the alternative Mg/Ca to temperature equation of (Elderfield and Ganssen, 2000) would only increase the temperature offset (Saher et al., 2009). An alternative explanation for the difference

in the alkenone Uk'37 and Mg/Ca based temperature records is changes in carbonate chemistry through the carbonate ion effect (Elderfield et al., 2006; Lea et al., 1999), which may affect the recording of temperature in the different proxy carriers (Saher et al., 2009). This problem is only important for Mg/Ca ratios in foraminifera at temperatures below ~3°C (Elderfield et al., 2006) and at pH values lower than present (Russell et al., 2004). In core NIOP 929 however, planktic foraminifera calcified at SST's well above 3°C (Rostek et al., 1997) and pH changes during the

Quaternary were likely in a range not significantly affecting Mg/Ca temperature estimates (Russell et al., 2004). It is therefore likely that the Mg/Ca records of core NIOP 929 have not been significantly affected by changes in carbonate chemistry through the carbonate ion effect (Saher et al., 2009). Another explanation is that as alkenone Uk'37 and Mg/Ca values are recorded by different organisms, differences in the seasonal or depth habitats between the two proxy carriers result in different SST estimates (Saher et al., 2009). The modern Arabian Sea alkenone

producing algae reflect annual average conditions (Prahl et al., 2000). Based on sediment trap data, the planktonic foraminifera *G. ruber* occurs in the modern Arabian Sea in roughly equal proportions during the winter and summer season (Conan and Brummer, 2000; Curry et al., 1992). This supports the notion that Mg/Ca- data based on sediments (which contain a mix of *G. ruber* from both seasons) reflect annual average conditions too. Differences between Mg/Ca and Uk'37 -based SST estimates may entail shifts of the seasonal or depth preference of *G. ruber* and/or the

alkenone producing species (Saher et al., 2009). An additional factor which may explain the higher alkenone Uk'37 temperature estimates in the 43 - 52 ka BP time window could be the preferential dissolution of high-Mg calcite in *G. ruber*, which would result in lower temperature estimates (Branson et al., 2013; Brown and Elderfield, 1996; Russell et al., 1994; Sadekov et al., 2005; Saher et al., 2009). Indicators of carbonate dissolution such as average shell weight, fragmentation, proportion of test loss during cleaning and CaCO3 content in the bulk sediment

(Tachikawa et al., 2008) are not available for the 37.4 ka BP to 67.7 ka BP section of core NIOP 929. Planktic foraminifera in the region, however, are generally well preserved at the depth of core NIOP 929 (Conan et al., 2002) supporting the notion that carbonate dissolution has not significantly affected our Mg/Ca data. Also, despite the small

areas of disagreement, the overall agreement between the two proxies across the studied period adds confidence to the *G. ruber* Mg/Ca SST record of this study.

The seasonal temperature reconstruction enables insights into the seasonal SST evolution in the western Arabian Sea. The outlined method does, however, assume a constant seasonal distribution of *G. ruber* and *G. bulloides* through time, which may not have been the case (Saher et al., 2007b). The seasonal abundance of *G. bulloides* reflects food availability by being present almost exclusively during the summer upwelling period (Conan and Brummer, 2000; Curry et al., 1992). In the past, however, during glacial periods with a weakened summer monsoon

and strengthened winter northeast monsoon (Anderson and Prell, 1993; Prell and Campo, 1986; Saher et al., 2007b), highest food availability may have shifted to the winter monsoon season. In addition, colder winter monsoons may have induced vertical mixing bringing nutrients to the surface (Reichart et al., 1998). The latter model has been proposed for the eastern Arabian Sea, but the change involved in such a scenario for the western Arabian Sea is currently unclear. The seasonal difference between the winter and summer monsoon SSTs of this study (Fig. 4, 7)

indicates strong summer monsoon winds generally persisting during the study period. This supports the notion that the glacial seasonal food distribution may have been broadly similar to the modern one, with the summer monsoon season driving the abundance of *G. bulloides*. Even if summer monsoon induced nutrient/food supply had dropped, during more extreme events, increased winter monsoon induced nutrient/food supply may not have occurred. This suggestion is based on results from the wider Arabian Sea showing that during the extreme Heinrich Events both,

SW and NE monsoons were weakened (Singh et al., 2011). This supports the notion that NE monsoon winds could not have driven nutrient/food supply in the western Arabian Sea even during extreme events such as Heinrich Events. This conclusion is in line with (Ganssen et al., 2011) in relation to HE's and on glacial-interglacial time scales with (Jung et al., 2002). Hence, it is unlikely that, during periods with lower temperatures prevailing during summer, the proportion of *G. bulloides* present during the winter northeast monsoon was substantially different than today. This

view is indeed in line with combined sediment core and model data from the northern Indian Ocean implying that the flux of *G. bulloides* is largely controlled by wind forcing (Bassinot et al., 2011), occurring during summer at the location of core NIOP 929.

Changes in the seasonal abundance of *G. ruber* are more difficult to assess. In the modern Arabian Sea close to site NIOP 929, *G. ruber* occurs during both seasons in roughly equal numbers (Conan and Brummer, 2000). This entails

that *G. ruber* is rather insensitive to changes in both, temperature, and food/nutrient availability, implying that it is able to tolerate changes in both parameters without significantly affecting its abundance. The data in Fig. 4 imply that the lowest and highest SST estimates of ~18 °C  and ~28-30 °C in both, the *G. ruber* and *G. bulloides* based SST records as well as in the seasonal time series,   broadly resemble the SST range in the modern Arabian Sea of

16°C (Swallow and Bruce, 1966) to 28.5 °C (Conan, 2006). These data are also in line with estimates of temperature

ranges recorded over the last ~20 ka in core NIOP 905 (Ganssen et al., 2011), close to the location of core NIOP 929. Based on temperature ranges between 37.4 ka BP to 67.7 ka BP in the western Arabian Sea being roughly similar to the modern range, *G. ruber* abundances may have been broadly stable as well, although additional research is required to assess the strength of this argument. We are aware that reconstructing monsoon variations at a seasonal scale is challenging, and some uncertainties are unavoidable. We do believe, however, that the data available for core

NIOP 929, are sufficiently robust to endeavour an initial assessment of seasonal change in monsoonal airflow in the Arabian Sea.

## 4.2.      Long-term monsoon variability in the Arabian Sea in relation to periods I to IV

A large seasonal SST difference between summer and winter temperatures stems from low (upwelling driven) summer SSTs and high winter SSTs. In Figure 4, SSTs during winter are generally higher than during summer. The

average difference in Fig. 4 for periods II and III, is around 8.5 °C. In periods I and IV the average seasonal SST differences are smaller. Broadly, this seasonal gradient is similar to the range in temperatures in the modern-day Arabian Sea (Swallow and Bruce, 1966). It is also comparable to the seasonal gradient of 5°C reported for the topmost section in core NIOP 929 using single foraminifera based SST estimates (Saher et al., 2007b). This supports the notion of a generally strong SW monsoon and a relatively weak NE monsoon prevailing between 37.4 ka BP to

67.7 ka BP, similar to the modern setup.

The definition of periods I-IV rests on absences/occurrences of periods with a reversed seasonal temperature gradient, with these reversals being short-term in nature. We note that within the uncertainty of the age model, there is no obvious relationship to known short-term events such as Heinrich Events (HE's; positioning of HE's in reference to the NGRIP record follows (Bond et al., 1999)), which applies to the *G. ruber* and *G. bulloides* based

SST-records as well. The  most pronounced temperature reversals in our seasonal temperature records at ~66.5 - 67 ka BP, ~65 ka BP, ~ 41.5 ka BP (with the caveat of this event being defined by one data point) and ~38 ka BP occur before or after HE's. These reversals are also recorded in the *G. ruber* and *G. bulloides* based SST-records, with the exception of the ~41.5 ka BP period, when the SST difference was only strongly reduced (Fig. 4). Also, there does not seem to be an obvious relationship with cold/warm oscillations in Greenland or Antarctic Ice cores (Fig. 4). This

finding indicates that pronounced changes in seasonal monsoon strength occurred without an obvious phase-locked relationship with the millennial-scale change patterns prevailing elsewhere. We will explore this relationship further below.

The short maxima in summer SSTs and minima in winter SSTs in core NIOP 929 imply a seasonal change in monsoon strength. Enhanced summer SSTs point to a reduction in SW monsoon airflow. Reduced winter SSTs indicate a

change during the NE monsoon season. It is difficult to pinpoint the specific cause of these cooling events. They could indicate cooler airmasses during the winter season affecting the Arabian Sea or alternatively, a stronger NE monsoon may have induced deeper mixing of the surface ocean allowing entrainment of cooler subsurface water into the mixed layer to be recorded in the planktic foraminifera record. A combination of both options is possible too (compare discussion in (Saher et al., 2007b).

Seasonal variation in monsoon strength has been reported in previous studies (Anderson and Prell, 1993; Prell and Campo, 1986), albeit with a focus on orbitally driven monsoonal change. Supporting findings at the millennial scale have been described for the last glacial period at 13 - 20 ka BP in core NIOP 929 (Saher et al., 2007a), showing rapid changes in both summer and winter SSTs. Also, (Anand et al., 2008) reports *G. bulloides* and *G. ruber* based SSTs from western Arabian Sea core NIOP 905 covering the last 35ka. Interestingly, there is a difference between *G.*

*bulloides* and *G. ruber* based SSTs between cores NIOP 905 (Anand et al., 2008) and NIOP 929 (this study). Core NIOP 905 does not seem to indicate a reversal of the temperature gradient between the species, but rather periods with a reduced gradient (Anand et al., 2008) in line with (Ganssen et al., 2011). This finding contrasts the data from core NIOP 929. Core NIOP 905 is located off the coast of Somalia, within the "Great Whirl", i.e. the central regional upwelling area. In the modern Arabian Sea, core NIOP 929 is affected by upwelling too, but to a lesser extent.

Summer upwelling indicative species *G. bulloides* occurred in core NIOP 905 over the last 35 ka BP even during Heinrich events (Jung et al., 2002), entailing coverage of full glacial conditions as well as superimposed short-term extremes, including a very strong Heinrich Event 1. The Heinrich Event-Dansgaard-Oeschger variability during the parts of MIS3 and MIS4 covered in this study probably does not comprise more extreme climate states than have occurred over the last 35 ka BP. This indicates that the SW monsoon winds varied in strength but persisted off the

coast of Somalia throughout the study interval of this study. Accordingly, weakened, but persistent SW monsoon winds would have suppressed a larger change in the seasonal SST gradient at site NIOP 905. At the location of core NIOP 929, during periods with a temperature reversal, the suppression effect, i.e. maintaining some SW monsoon induced upwelling, would be smaller, possibly along with a reduction of the size of the Great Whirl. This in-turn supports the notion of summer upwelling being more susceptible to disruptions, entailing larger SST changes at site

NIOP 929, in line with the large swings in temperature in our data.

Similar to core NIOP 929, the time series for core SK17 from the eastern Arabian Sea (Anand et al., 2008) show short term reversals in the *G. bulloides* and *G. ruber* based SSTs over the last 35 ka BP. Off the west coast of India, periods of food/nutrient enhancement in surface waters, which *G. bulloides* responds to with high abundances

(sensu,(Curry et al., 1992), may occur during both winter and summer. A small amount of localised upwelling occurs during summer with similar conditions sporadically developing during winter (Cullen and Prell, 1984). In either season, the upwelling is weak compared to the strong upwelling in the western Arabian Sea off Somalia. Without persistent strong upwelling it is probable that short-term disruptions of the monsoon circulation during MIS3/4, reflected in the temperature reversals in core NIOP 929, affected the eastern Arabian Sea in a similar way. Rather than involving a subtle change, monsoon circulation was more substantially affected, leading to reversed seasonal temperature gradients in the eastern Arabian Sea (sensu (Anand et al., 2008)). Principal support for the notion of large changes in monsoonal circulation during MIS3 comes from the west Pacific warm pool (Stott et al., 2002), north China (Wang et al., 2021) and the Japan Sea (Sagawa et al., 2023), suggesting that the wider Asian monsoon circulation displayed significant variability during MIS 3/4.

### 4.3.    Phasing of seasonal millennial scale SST variations in the Arabian Sea and beyond

One of the most striking   findings of our study is the presence of short-term positive and negative temperature excursions in core NIOP 929 (Fig. 4). Whilst other proxy records from the Arabian Sea (Altabet et al., 1995; Ivanochko et al., 2005; Schulz et al., 1998) revealed a systematic relationship with glacial northern hemisphere millennial climate change, our SST records do not display such a phase-locked persistent relationship. During HE6 for example in Fig. 4 the *G. bulloides* and *G. ruber* based SSTs as well the summer SST record indicate cooler conditions. During HE4 in contrast, the *G. ruber* based SST record indicates a warming event. Both, the *G. bulloides* based SST and the summer temperature time series, however, show a short-term SST reduction embedded in an overall temperature rise during HE4. Comparable observations from the Arabian Sea are in (Anand et al., 2008) and to some degree in (Saher et al., 2007b) supporting  the notion that temperature changes in the region are not phase locked and consistent with millennial scale climate oscillations in the northern hemisphere. Although there are a range of possible explanations involving local/regional changes in water properties (see discussions in (Anand et al., 2008; Saher et al., 2007b)), the data from core NIOP 929 may offer an alternative view. This notion is based on the finding that some short-term events in our  summer monsoon affected SST records (*G. bulloides* SSTs and the derived summer SST's), coincide with short-term changes in the Antarctic ice core $\delta^{18}$O record in Fig. 4. It is not a perfect correlation, but within the uncertainty of the age models it does indicate that some of the temperature changes in core NIOP 929 do reflect a link with southern hemisphere climate change, being most obvious in the top part of our study interval. Assessing the seasonal large-scale meteorological change driving monsoon circulation in the Arabian Sea, allows taking this notion further. It could be argued that the seasonal shift in the location of the Inter Tropical

Convergence Zone (ITCZ) entails a switch in hemispheric "climate dominance" in the Arabian Sea. The ITCZ, the thermal equator, could be regarded as the dividing line/area, with northern hemisphere influence dominating north of it and southern hemispheric influence governing to the south. Within each meteorological hemisphere, polar climate changes could be transmitted relatively easily equatorward, with the ITCZ constraining further propagation. In view of the concept of the bipolar seesaw, i.e. phase-offset cold warm oscillations between the poles during MIS2-5 (Epica, 2006), this might shed light on the rather complex relationship of millennial scale temperature changes in the Arabian Sea with related change elsewhere. During northern hemisphere summer, the ITCZ reaches its northernmost position, i.e. being located over the Asian continent. During this season, southern hemisphere climate conditions would affect the Arabian Sea. Hence, millennial scale oscillations recorded over Antarctica (Epica, 2006) and references therein) could have affected monsoon circulation in the Arabian Sea chronicled as SSTs changes broadly following the southern hemisphere signal. During northern hemisphere winter, the ITCZ resides south of the equator, allowing northern hemisphere climate variations driving the monsoonal record in the Arabian Sea. This scenario entails that the Arabian Sea potentially records both northern and southern hemisphere millennial-scale change patterns. A non-straightforward and not phase-locked relation of the millennial scale SST records from the Arabian Sea with similar change elsewhere would result from this, as indicated in Fig. 4. Such a scenario would also explain why the winter SST record displays the least similarity with the southern hemisphere and its long-term change resembles insolation change at 65°N, suggesting a close relation with the north. This scenario is in line with a growing body of research challenging the paradigm of a solely northern hemispheric control on monsoon variability on millennial to orbital time scales (An et al., 2011; Caley et al., 2013; Rohling et al., 2009). In particular the comparison of the Hulu Cave data from continental China and ice core data from Antarctica (Rohling et al., 2009) and sediment data from the Arabian Sea (Caley et al., 2013) promote the idea of a stronger southern hemisphere monsoon control in relation to millennial scale change during glacial periods. Our data add to this idea by supporting the notion that, periodically, the southern hemispheric influence is mainly exerted during northern hemispheric summer, similar to earlier suggestions (Wang et al., 2013).

## 5.    Conclusions

The temperature reconstruction of core NIOP 929 using Mg/Ca paleothermometry of *G. ruber* and *G. bulloides* shows a range of 9°C in both records and rapid temperature changes from 37.4 ka BP to 67.7 ka BP. The novel approach of using the Mg/Ca SST records of *G. ruber* and *G. bulloides* to reconstruct winter and summer monsoon SSTs allows assessment of monsoonal controls at a seasonal scale. The results indicate that the summer southwest

monsoon was strong for most of the period from 37.4 ka BP to 67.7 ka BP but was significantly weakened multiple times, reversing the normal seasonal temperature gradient in the region. Also, the SST records do not show the expected phase-locked millennial-scale relationship with change in Greenland ice cores, nor do they show such a

relationship with southern hemisphere millennial-scale climate change. The supports the notion of the Arabian Sea being a melting pot of climate controls, with both northern and southern hemisphere influence being recorded in sediments from the region. Further research is required, using a broader range of climate proxies such as productivity records, to verify the main implications of our study.

## 6.      Author contribution

All authors contributed to the data used in the study, with JS and DC adding the Mg/Ca-data and JS providing the stable isotope data. JS and SJ led the writing process with contributions from DC.

## 7.      Competing Interests

S. Jung is co-editor of the special issue dedicated to Dick Kroon and will not be involved in the handling of this manuscript. The other authors declare that they have no conflict of interest.

## 8.      Acknowledgements

We would like to thank Dr Laetitia Pichevin at the School of Geosciences of the University of Edinburgh for helping with the analytical side of the project. We are grateful for numerous final year students for helping to establish the stable isotope records for core NIOP 929. We are also grateful for support of this project by colleagues at Heriot Watt University - Dr Babette Hoogakker and Dr Katrina Nilsson-Kerr and to the members of the Heriot Watt Science

Discussions group as well as to Dr Kate Darling for her insightful thoughts on foraminifera ecology.

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

## 10. Figure captions and table headings

| Depth (cm) | Original age from Saher et al., (2007) (kaBP) | Recalibrated age - median (kaBP) | 1 sigma age range (kaBP) | 2 sigma age range (kaBP) | Recalibrated - original age (kaBP) |
|---|---|---|---|---|---|
| 21.75 | 1.292 | 1.305 | 1.197 - 1.398 | 1.102 - 1.516 | 0.013 |
| 50.75 | 5.699 | 5.693 | 5.579 - 5.804 | 5.475 - 5.905 | -0.006 |
| 76.75 | 8.929 | 8.928 | 8.780 - 9.069 | 8.632 - 9.209 | -0.001 |
| 82.25 | 9.737 | 9.779 | 9.603 - 9.912 | 9.526 - 10.088 | 0.042 |
| 125.25 | 13.508 | 13.569 | 13.449 - 13.700 | 13.312 - 13.805 | 0.061 |
| 180.25 | 18.313 | 18.031 | 17.901 - 18.177 | 17.712 - 18.289 | -0.282 |
| 210.25 | 19.92 | 20.035 | 19.865 - 20.212 | 19.663 - 20.375 | 0.115 |

Table 1: Original and updated ages from Saher et al., (2007) 14C ages for core NIOP 929, recalibrated to calendar ages using the 14C age calibration program Calib 8.2 (Stuiver and Reimer, 1993), using a ΔR value of 45 (±67)

(Southon et al., 2002).


| Depth (cm) | Age (kaBP) |
|---|---|
| 317.5 | 31 |
| 496.5 | 45 |
| 669.5 | 62 |
| 832.5 | 87 |
| 958 | 107 |
| 1025.5 | 126 |

Table 2: Age control points based on tuning to the LR04 $\delta^{18}O$ global stack of Lisiecki and Raymo (2005).

| depth (cm) | age (initial LR tuning; kaBP) | new age based on NGRIP tuning (kaBP) | note |
|---|---|---|---|
| 251 | 24.13 | 23.14 | NGRIP based age |
| 317.5 | 31 | | LR04 based control point used |
| 357.5 | 34.128 | 35.2 | NGRIP based age |
| 396.5 | 37.179 | 37.92 | NGRIP based age |
| 463.5 | 42.419 | 41.86 | NGRIP based age |
| 496.5 | 45 | | LR04 based control point used |
| 571.5 | 52.37 | 53.88 | NGRIP based age |
| 669.5 | 62 | | LR04 based control point used |
| 671.5 | 62.307 | 62.9 | NGRIP based age |
| 727.5 | 70.896 | 70.02 | NGRIP based age |
| 741.5 | 73.043 | 73.18 | NGRIP based age |
| 774.5 | 78.104 | 77.28 | NGRIP based age |
| 832.5 | 87 | | LR04 based control point used |
| 958 | 107 | 109.52 | LR04 control point with revised NGRIP based age |
| 1025.5 | 126 | | LR04 based control point used |

Table 3: Combined age control points based on tuning to the LR04 $\delta^{18}O$ global stack of Lisiecki and Raymo (2005)
and the Greenland ice core NGRIP $\delta^{18}O$ record (Andersen et al., 2006; Rasmussen et al., 2006; Svensson et al., 2006; Vinther et al., 2006; Wolff et al., 2010).



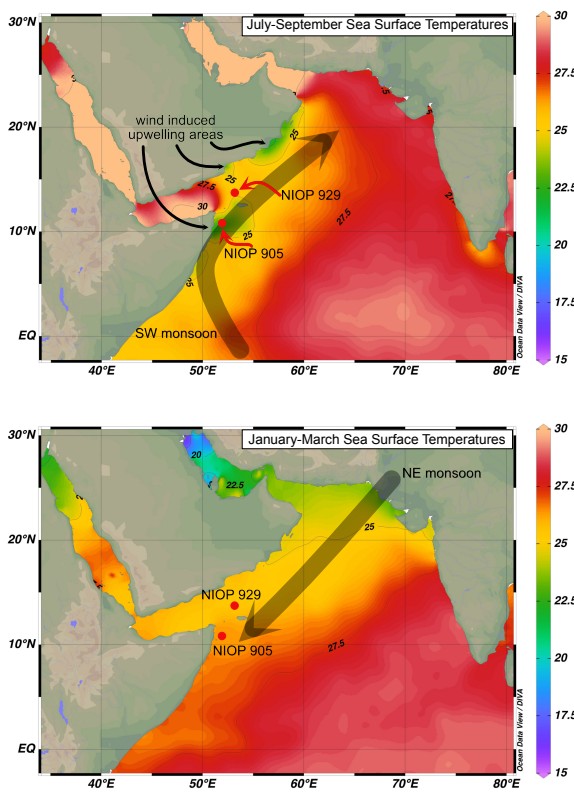

Figure 1: Maps of the Arabian Sea region. The locations of core NIOP 905, NIOP 929 are indicated. Sea surface temperatures for the summer season (July-September) and the winter season (January-March) are based on the World Ocean Data base 2023 (Reagan et al., 2024). The base graphs were created in Ocean Data View. Black arrows indicate

rough positions of summer upwelling areas. Transparent arrows indicate prevailing seasonal airflow.


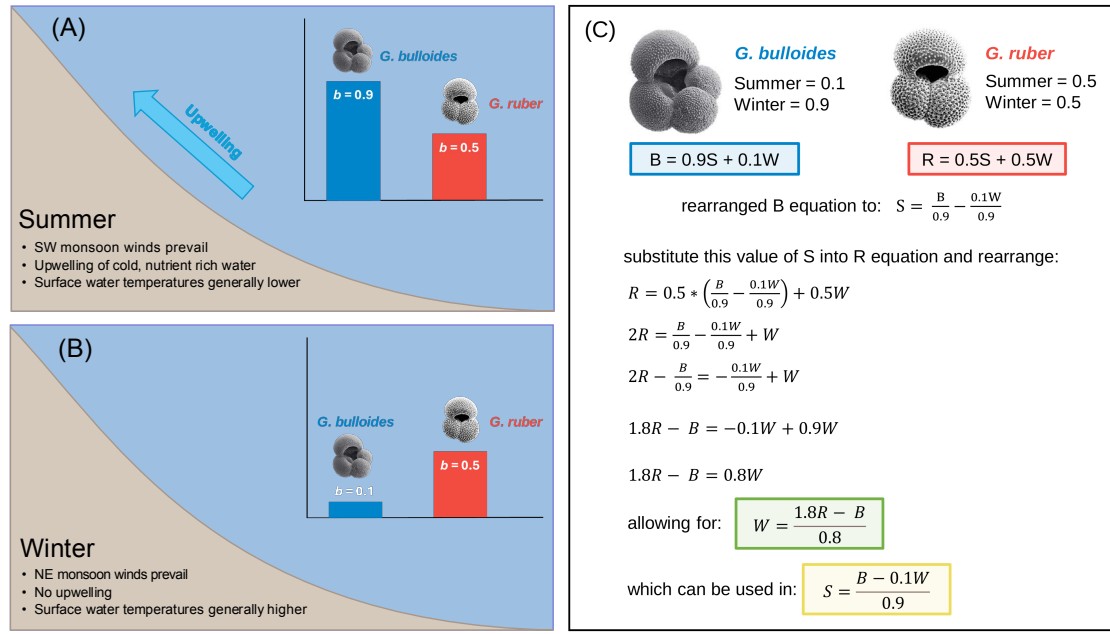

Figure 2 Seasonal foraminiferal flux dynamics in the western Arabian Sea for *G. bulloides* and *G. ruber*. (a) Typical summer conditions and (b) typical winter conditions. (c) Winter and summer temperature estimates derived from Mg/Ca temperature estimates from *G. bulloides* and *G. ruber* and seasonal foraminiferal flux dynamics.




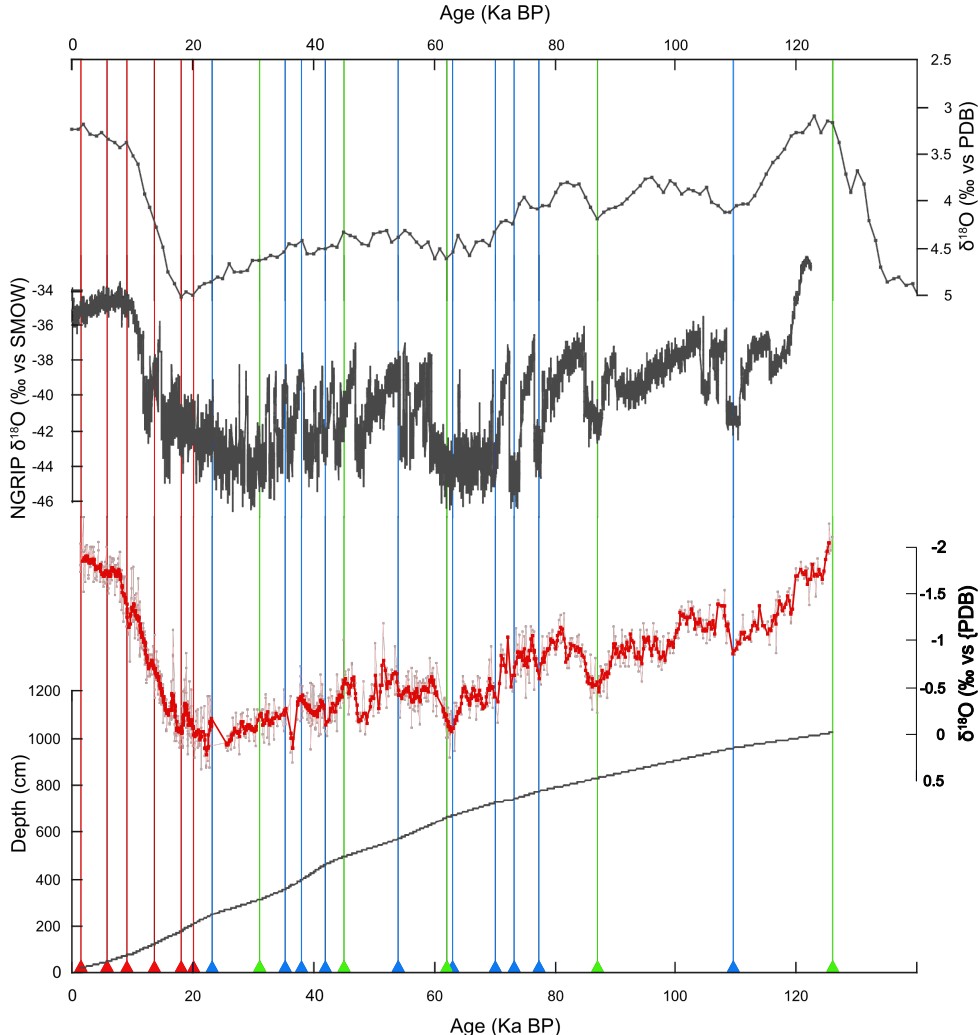

Figure 3: Overview of the age model development of core NIOP 929. The records shown are (top down) the LR04 δ[18]O global stack of Lisiecki and Raymo (2005), the Greenland ice core NGRIP δ[18]O record (Andersen et al., 2006; Rasmussen et al., 2006; Svensson et al., 2006; Vinther et al., 2006; Wolff et al., 2010), the δ[18]O- record *G. ruber* of core NIOP 929 (700 year boxcar filtered record in bold, original record at reduced opacity) and the depth-age record for core NIOP 929. Pairs of triangles and vertical lines indicate origin of age control points; red = AMS[14]C, green =. LR04 and blue = NGRIP. For uncertainties see text.

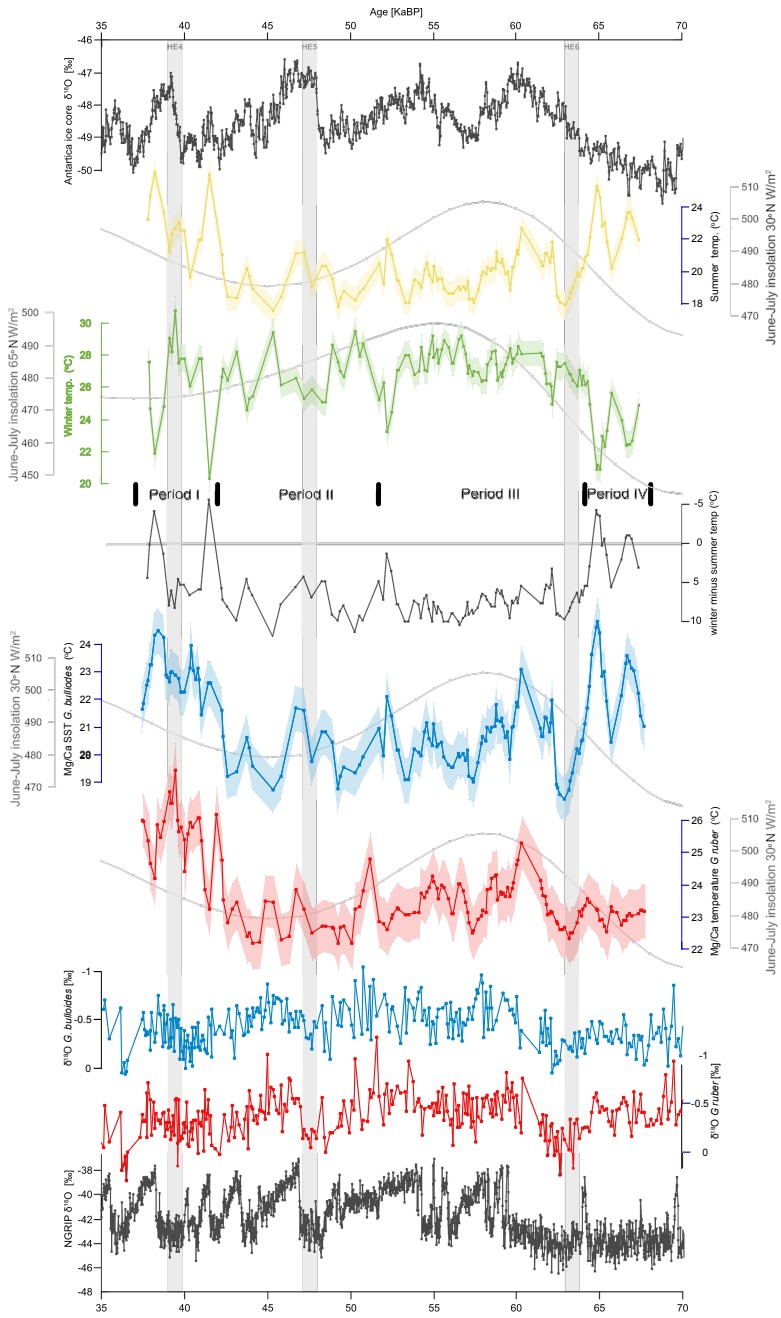

Figure 4: Summary figure showing (top down): the Antarctic EPICA $\delta^{18}$O record (Veres et al, 2013), summer temperature estimates (this work), winter temperature estimates (this work), difference between winter and summer SST estimates (winter minus summer; grey line depicts temperature equivalence, negative values indicate temperature reversals), SST estimates based on Mg/Ca data from *G. bulloides* (this work), SST estimates based on Mg/Ca data from *G. ruber* (this work), stable isotope values for *G. bulloides* and *G. ruber* (this work), and the

Greenland ice core NGRIP $\delta^{18}$O record (Andersen et al., 2006; Rasmussen et al., 2006; Svensson et al., 2006; Vinther et al., 2006; Wolff et al., 2010). Grey bars indicate position of Heinrich Events (HE's). A 700 year box car filter has been applied to all SST timeseries. In order to indicate uncertainties in our data set, error envelopes have been added to all SST reconstructions. For details, please see main text.

    Periods I-IV, discussed in the text, are indicated. Positioning of HE's in reference to NGRIP follows Bond et al.

(1999). Insolation data taken from Laskar et al. (2011).





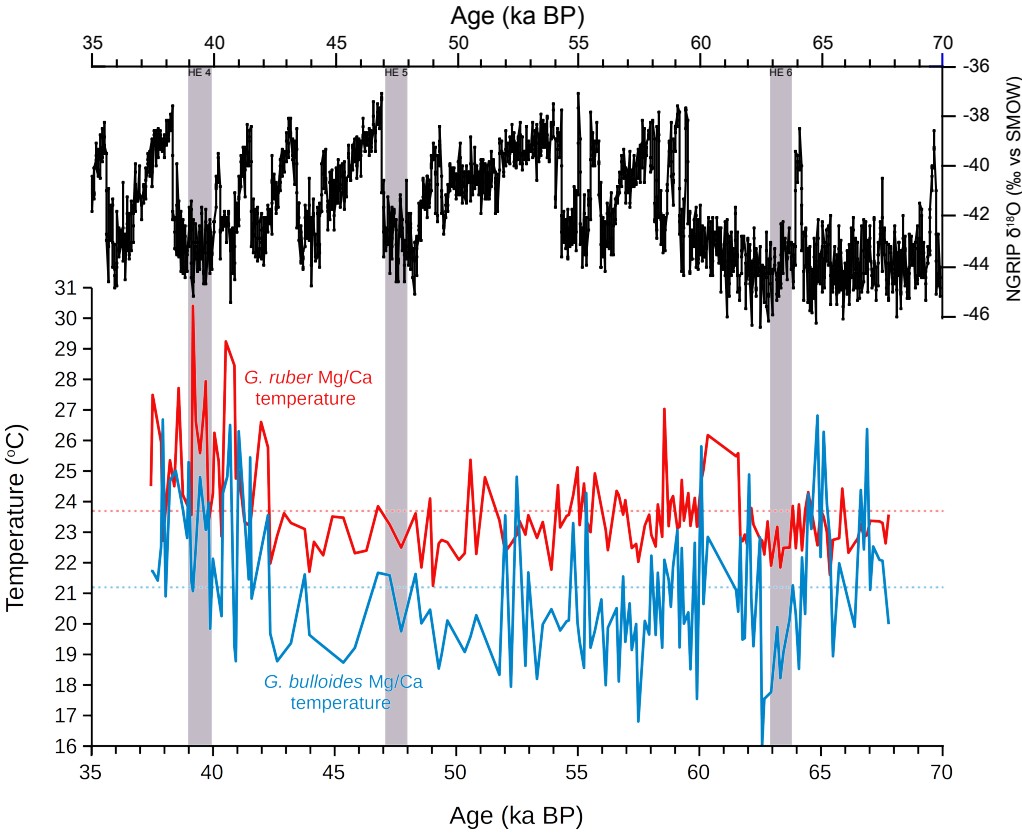

Figure 5: Mg/Ca based temperature reconstructions for *G. ruber* (red) and *G. bulloides* (blue) from core NIOP 929. Dashed lines in the temperature records indicate the mean temperature values for each record across the reconstructed period.



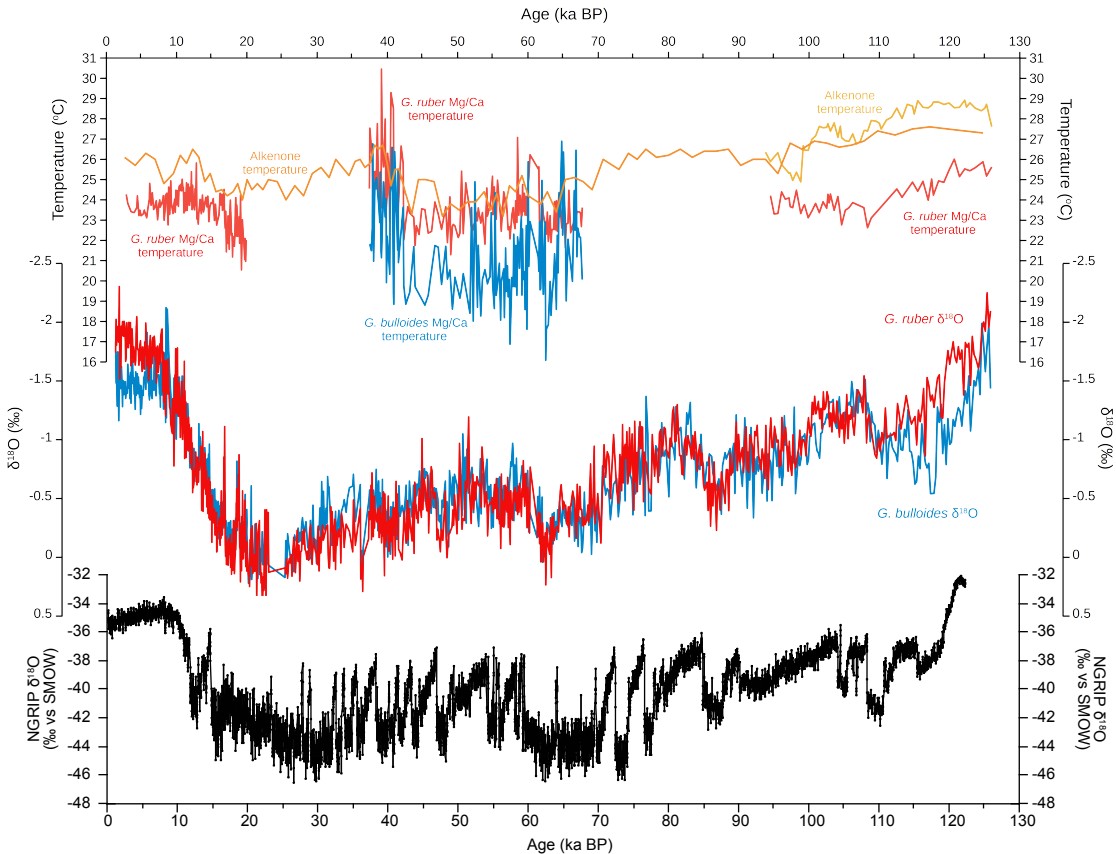

Figure 6: Temperature history of core NIOP 929 to 126 ka BP including Mg/Ca temperatures, alkenone temperatures and stable oxygen isotope values. All records are placed on the age model created in this study. *G. ruber* Mg/Ca temperature records of this study from 37.4-67.7 ka BP, Saher et al., (2007) from 2.9-20 ka BP and Saher et al., (2009) from 95.5-124 ka BP in red. *G. bulloides* Mg/Ca temperature record of this study from 37.4-67.7 ka BP in blue. Alkenone temperature records of Rostek et al., (1997) in dark orange and Saher et al., (2009) in light orange.

*G. ruber* $\delta^{18}$O record of this study in red and *G. bulloides* $\delta^{18}$O record of this study in blue. NGRIP $\delta^{18}$O record in black.


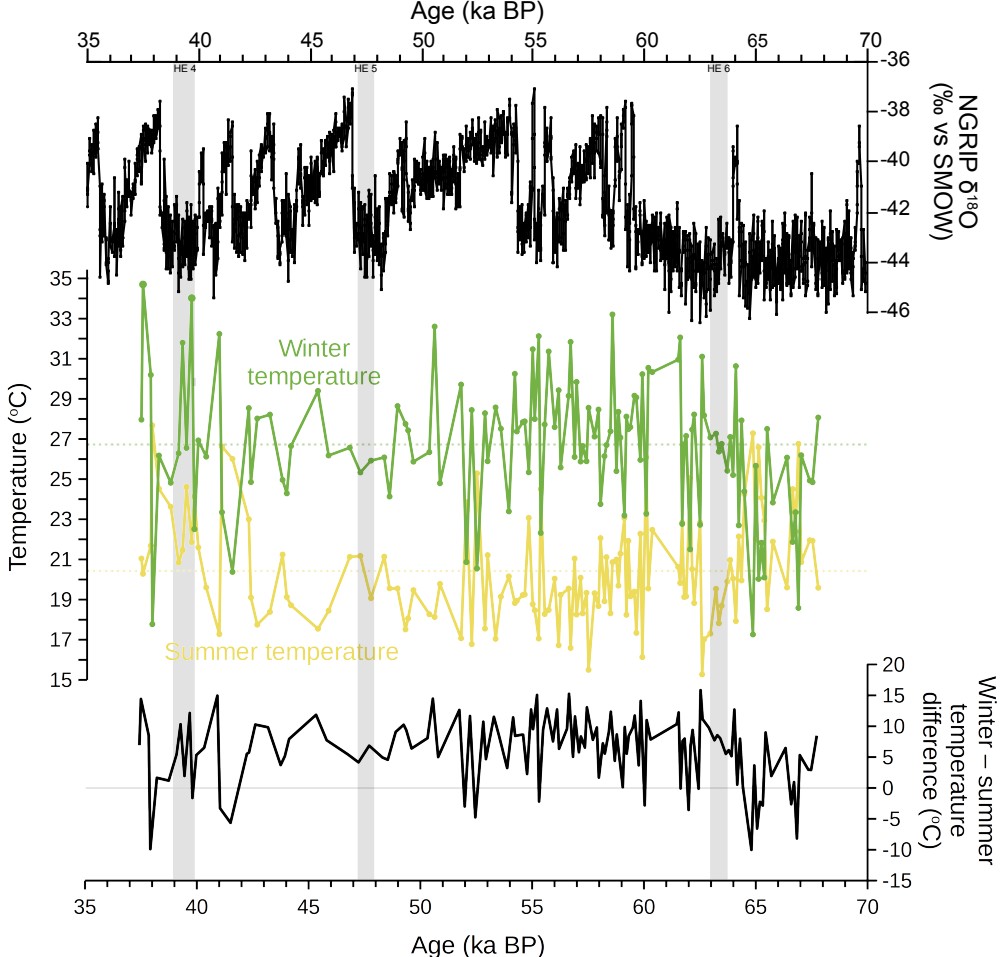

Figure 7: Winter and summer temperature estimates from core NIOP 929. From top down, NGRIP $\delta^{18}O$ record (black), winter (green) and summer (yellow) temperature reconstructions from core NIOP 929 and the difference between winter and summer temperatures (black). Dashed lines in the temperature records indicate the mean temperature values for each record across the reconstructed period.