# Peer review of "Millennial scale sea surface temperatures of the western Arabian Sea between 37 - 67 ka BP"

_EGUsphere, 2024_

## Author Response (AR1)

**Author responses to Anonymous Referees for "Millennial scale sea surface temperatures of the western Arabian Sea between 37 - 67 ka BP"**

https://doi.org/10.5194/egusphere-2024-865

For ease we have copied the referee comments above the author responses.

In case where our response is in line with our earlier response, the answer is color coded in green. In case, we have reconsidered our response, it is color coded in red.

**RC1 - https://doi.org/10.5194/egusphere-2024-865-RC1**

*"The manuscript "Millennial-scale sea surface temperatures of the western Arabian Sea between 37–67 ka BP" by Scott et al. presented excellent new data of paired Mag/Ca SST data from the Arabian Sea. The data covers a critical time in the North Hemisphere that experiences abrupt oscillations of millennial-centennial scales; therefore, it is valuable to test any linkage or temporal or phase relationships between the high and low latitudes climate. The data are well-presented, and interpretations are reasonable and thus publishable in EGUsphere. Though the manuscript is worthy of publication, I suggest the authors show the age model uncertainty of the core (Figure 3) while comparing it with the ice core. Several statistical methods allow the authors to show the C14-based age model with age uncertainties associated with the dating. The revised figures with age uncertainties will be more helpful in examining the temporal relationships between the H events and rapid changes in the Arabian monsoon, and the interpretation could be better incorporated into the text of the next version. The H events should be highlighted in Figure 6, too, and also need to be examined for any temporal relationships between the NH cooling and temperature gradient (upwelling?) in the Arabian Sea. I will be more than happy to evaluate the following revised version of the manuscript and look forward to the publication of the paper."*

**Response to RC1**

We thank Referee 1 for their valuable comments and their time spent reviewing our manuscript. We thank them for the positive feedback and insightful comments. We will add an improved age modelling chapter to the next version of the manuscript which will include additional details related to the age model generation as well an assessment of uncertainties. We will also revise our figures to ensure temporal relationships between different records (including timing of Heinrich Events) are easier to compare.

**RC2** - https://doi.org/10.5194/egusphere-2024-865-RC2

*The paper re-visits the classical sediment core NIOP 929 in the western Arabian Sea that has been previously used for studying surface ocean and paleoproductivity changes in response to variations in monsoon circulation. The paper extends previous data by adding high resolution Mg/Ca based sea surface temperature records (based on G. bulloides and G. ruber) for marine isotope stage 2 to 4, covering the time-interval of large-scale millennial-scale climate fluctuations as known for example from ice-cores.*

*The paired Mg/Ca-based SST records allow to distinguish temperatures during both modern winter and summer monsoon seasons with high temporal resolution, clearly sufficient to display the pastern and timing of millennial-scale changes. Both SST records reveal high amplitude fluctuations, being much larger than millennial-scale SST changes during the last deglaciations or the previous interglacial. Importantly, these summer and winter SST changes do not show a systematic relationship with glacial northern hemisphere millennial climate change. This is important as the Arabian Sea and the northern hemisphere monsoon have been traditionally considered being primarily connected with the Greenland millennial-scale pattern. The authors discuss possible scenarios, also involving seasonal shifts of the ITCZ with potential more southern (Antarctic) influence during boreal summer when the convergence zone moves northward.*

*Overall, the paper is carefully written and the discussions are detailed and nicely involve previous findings form paleo-records in the Arabian Sea. It is hard to objectively assess the timing and pattern of the SST records in terms of northern versus southern millennial-scale timing. I partly agree with the comments of reviewer RC1 regarding a better presentation of the age model. However, radiocarbon dating will probably not provide sufficiently small errors to unequivocally connect to Greenland versus Antarctic pattern as 37-67 ka is at the limit of 14C dating. Therefore, the tuning approach with d18O records is probably the best age control possible for the older part of the record.*

*As quite some interpretation relies on the visual comparison to the ice-core records, I suggest to include them in all figures 4 to 6 to allow better illustrate the suggested non-straightforward relation of the summer and winter records to northern and southern hemisphere climate changes at millennial time-scales.*

**Response to RC2**

We thank Referee 2 for their valuable comments and their time spent reviewing our manuscript. We are grateful for the positive feedback on our work and the insightful comments on how to improve our manuscript. As also touched on by Referee 1 we agree with Referee 2 that improved design of Figures 4-6 will allow for easier comparison between records. We will integrate this change into our revised manuscript. Please also see response to RC1.

**RC3** – https://doi.org/10.5194/egusphere-2024-865-RC3

*Sorry for this long review. I really enjoyed the article, but there are several critical problems. Perhaps some of my comments can't be dealt with, and that's completely OK but please explain why you don't agree with my comment.*

*The article presents a high-resolution, high-quality SST and isotopic data from a marine sediment core collected offshore Somalia. The new data focus on MIS3, and are discussed along previously published data from the same core (but focusing on different time intervals) and other cores collected in the region. The analysis attempts resolving winter and summer SST variability using Mg/Ca measurements performed on planktonic foraminifera with contrasting seasonalities. And that's a difficult task - so I commend the attempt. The authors conclude that the ITCZ latitudinal movements have differential impact on the overall SST signals w.r.t. the hemisphere that controls the signal (northern vs. southern hemisphere in boreal winter vs. summer, respectively) that alter the possibility to read a clear northern vs. southern signal in the seasonal SST regimes reconstructed using both foraminifera species.*

*I think the article and its interpretation should eventually be published, and that EGUSphere is a perfect journal to publish in. However I think a deep and thorough revision needs to be done before.*

*First, in a region where such a complex seasonality (summer being colder than winter) without a description of modern climatology in a full chapter is simply not possible. Figure 1 is not helpful in that regard. Please add a full paragraph and figures to describe clearly it.*

*Second, a deep and thorough description of the age model, with the mandatory uncertainties and an appropriate and fair discussion on the likely very much larger uncertainties for MIS3 has to be added. There not even mention of the s.d. of the radiocarbon dates... In particular in articles dealing with MIS3 the uncertainties of the age w.r.t. the ones applying to ice cores are just so much different that you can't hide this limitation.*

*Third, the way you present the winter and summer equations have to be fully described in the article. It is not possible to simply refer to other articles. The reader has to have in hands a minimum of background, and certainly with an extra figure showcasing the seasonality in G. ruber and bulloides to evaluate the winter and summer equations without downloading a series of articles.*

*I really think those three items need to be seriously dealt with prior to any attempt to discuss the science.*

**Response to general comments in RC3**

We thank Referee 3 for their valuable comments and time spent reviewing our manuscript. We thank them for the detailed feedback and insightful comments.

We agree with Referee 3 that a more in depth description of the modern climatology (including air sea interaction would be helpful. Our manuscript we will be updated accordingly. As touched on by all referees we will better describe the uncertainty of our age model, and as

suggested by Referee 3 ensure we describe the age model methodology and robustness in more detail. Additionally, we will include a more detailed description of the winter and summer temperature equations and the foraminifera flux dynamics data that this is based on to ensure that the methodology we have used is clear and accessible to the reader.

*This being said, I now start the formal review in the order it appears directly in the article.*

*2.3. (Age model): I don't think the authors can justify a fine-tuning of d18O to NGRIP as long as they conclude that SST don't look to any of the ice cores. Please discuss a bit more this processing. Also, please show the original d18O values and the result of the filtering on the same figure and Y axis. It is unclear which dataset is already processed.*

Thanks for these suggestions. We will show original d18O values before filtering and make clear in the figures what data is being shown (filtering on it etc). We will also address the reasons for our tuning approach in the revised age modelling chapter. Done.

*2.5. (W and S SST): I find the r and b values strange and arbitrary in Saher. Please discuss this issue here. In general, I don't like the way to assign some weight in the W equation with both species and the W into the S SST estimation equation. It doesn't make any mathematical sense to me. Playing around with this set of equations could drive the math towards uncomfortable solutions. For example, very small changes in the numbers of the parameters lead me to estimate S SST uniquely with ruber... but it is mean-annual. Also, the S SST as it stands depends partly on the W SST, which doesn't make any sense to me either, etc. So, please, even if I won't agree with your interpretation in the end (I'm fine with that, and will accept your way of interpreting), at least please elaborate more discussion on the choice of your parameters because the resulting W and S SST depend crucially on the exact value of these parameters.*

We will include a more elaborate assessment of the seasonal *G. ruber* and *G. bulloides* flux values from Saher and the choices we made in the winter and summer temperature estimates. Done.

*3.1. (d18O): in fact, I'm not sure that I understand it well: are the d18O are smoothed or not? Please show first the raw values.*

This will be clarified.

*4.1. (robustness…): please also make sure in the text and interpretation of different species and Uk'37 to be as firm as possible. You sometime state that Mg/Ca G. ruber and Uk'37 represent mean-annual SST. Then you may also elaborate why this might not be valid anymore during the MIS5, that by the way implies the season you assign to different proxies are not constant over time, that adds degrees of freedom in the interpretation to the seasonal SST changes and associated mechanisms described.*

We have re-read our text again and am unsure how we can include the essence of this comment. We have touched on a number of reasons why the alkenone and Mg/Ca based SST estimates might differ, including the seasonality aspect. We do therefore consider this point being covered. As for the comment in relation to MIS5, our work mainly focuses on MIS 3 and MIS4. We are therefore not quite sure how this comment applies to our work.

*Line 210: if you have the SST estimation, a very simple calculation given the calibration equation used in Rostek will give the Uk'37 you want to calculate to investigate different calibrations. So yes, that is a very basic option.*

Thanks for this suggestion, which we will consider during the revision of the manuscript. The main thrust of this manuscript is, however, to use the "seasonality" of the abundance of planktic foraminifera species to reconstruct seasonal monsoon change. Assessing different Uk'37 calibrations for data that are not our own may distract from our main aim.

*Sentence lines 216-218: I don't understand your point here.*

Section is rewritten.

*Sentence lines 222-224: Again, a specific paragraph is absolutely needed to describe the different foram fluxes you cite here.*

The is clarified in the text.

*Line 230: it is curious: there is no foram weighting prior to foram dissolution before analyzing Mg/Ca? This should have been a very basic/classic data reported somewhere.*

Previous studies (e.g. Saher 2007) have successfully used Mg/Ca thermometry in sediments from the Arabian Sea without the use of foram weights. This approach is supported by carbonate preservation studies in the region (Conan 2002). Whilst being interesting to measure foram weights, we are not sure that these data are essential for our data set. We have amended the wording in this section.

*Sentence lines 232-234: I don't agree with this statement. In this region, alkenones often/always provide a point-to-point scattering that is much lower than Mg/Ca for many reasons even when you incrase resolution.*

The sentence has been removed.

*Discussions lines 244-262: please try to structure the statements, it goes a bit in any direction. For example, again, if G. ruber tolerates everything, stil its Mg/Ca fluctuates. The fact that it is mean-annual has repercussions on the meaning of your seasonal index I criticized earlier. I also feel a larger discussion of the Ganssen data (on individual foraminifera analysis from the same species) should be detailed very much more, there is some important take-home message for you in their analysis on seasonality.*

We are thankful for this comment to bring in individual foraminifera analysis (Ganssen's work) when discussing seasonality data. This has been done.

*4.2. (long-term…): dealing with H4, well… please be clearer on the limits of your age model. Clearly, the H4 grey bar on the NGRIP could be larger and finish right on the following D/O event, while the corresponding grey bar width on the d18O of ruber is… unknown. Specifically for H4 it has been already described that the H4 has two distinctive events, so the grey bar width could perhaps be expanded very much, having quite an implication on your discussion on the seasonality you describe for your site. Then, dealing with the SW/NE monsoon evolution, interesting model simulations available for the Holocene in Bassinot et al., 2011, Climate of the Past, could help describing the shifts in foramfluxes for different species under varying climate conditions.*

In relation to the double spike related to HE4, please see response to comment on section 4.3 below.

In relation to the Bassinot work, this has now been included in the text.

*Sentence lines 289-290: again, you may cite Ganssen and discuss his data. Also, the fact that Uk'37is even warmer than G. ruber might reveal in fact that Uk'37 is NOT the mean-annual SST.*

The Ganssen work has now been included. Please see earlier comment of Uk37.

*4.3. (phase of…): again in the Naughton (EPSL, 2009) article there is clear 'double H4' signature. It may allow you enlarging your sedimentary sequence thickness of your own H4 and changes your interpretation.*

We are not sure that we agree with the Naughton work being relevant for our paper in terms of the definition of HE4. Naughton work is mainly based on a sediment core from the Portuguese continental margin. Naughton et al. states that the IRD record may not reflect the Heinrich Events as well as in areas further north. In order to better constrain the positioning of the HE's alternative indicators are used, deviating from the strict definition of a HE, i.e. being an ice rafting event. Hence we are not convinced that the definition of Naughton is helpful when determining the position of HE4. This is supported by data from sediment cores within the main IRD deposition zone, not showing a clear sign of a HE4 double spike (Hemming 2004)

*The ITCZ mechanism described is interesting, but already described in length in the western tropical Indian Ocean using G. ruber and Uk'37, model simulations etc. (see Wang et al., 2013, Paleoceanography). You really have to cite that article of you keep the discussion as it is (that I liked).*

Reference has been included.

---

## Author Response (AR2)

Referee and Editor comments. (*responses are in italics*)

The article by Stott et al. has been revised and many points raised by my and the two other referees have been addressed.

Yet I feel with a little more efforts the article could still be substantially improved.

*We thank the referee for the careful assessment of the revised version of our manuscript. Below, we respond to the issues raised.*

In particular, on the first figures presenting the modern conditions (Figure 1) could really be improved. There are now many tools/ways to provide information on the seasonal/subseasonal SST changes along the year, that could very much help visualize the description you made of the regional settings, that could also have been put in a proper paragraph (and not in the introduction).

*We thank the reviewer for this comment. Figure 1 has been substantially redesigned so that it now shows seasonal surface ocean temperature distributions in the Arabian Sea alongside prevailing airflow.*

I also admit I still don't see much of a point to work around complex equations for winter vs. summer SST, and would rather opt for having a more straightforward bulloides for winter and ruber for summer, as the fluxes reported in the Conan and Brummer foram flux reference shows that one replaces the other on a 'quasi systematic' way… so I have to say your description of the equations is still obscure to me.

*We agree that SST-estimates that are based on a single foraminifera species are a robust tool. In the previous version of the manuscript, we did use single foraminifera based SST's and did not solely rely on our seasonal SST. Having said this, monsoon circulation it the western Arabian Sea is, per definition, highly seasonal. Of the two species we used, only G. bulloides qualifies as an almost seasonal signal carrier given its near exclusive growth during the summer season.  This is reflected in the highly similar SST estimates based on G. bulloides and the summer temperatures in figure 4. This figure also shows that the G. bulloides and the G. ruber based SST's show some similarities, but also significant differences. Rather than just discussing both records individually as well as their similarities/differences in a generic seasonal monsoon context, our aim was to go deeper by exploring the seasonal temperature signal using our (we believe) rather novel approach. Given the availability of the data required for the calculations,  it would be a missed opportunity to not use these data sets. As with most new techniques, there are uncertainties related to our approach as well, i.e. the lack of data constraining changes in seasonal preference of either species. We have been discussing this in the manuscript. We therefore feel that, whilst not being perfect, the seasonal SST data add an aspect to variability in monsoonal dynamics that would otherwise be missed. In order to ensure not overinterpreting our SST data we have made small corrections to the manuscript,*

*highlighting when the single specimen-based G. bulloides and G. ruber SST records reflect variability seen in our seasonal SST estimates.*

On your figure 6, what would be the summer SST based on G. ruber? Would it increase the matching with alkenones? Or are there other potential mechanisms that could explain why alkenone look warmer than the summer SST derived from you equations?

*We are not quite sure what is meant here. Our seasonal SST reconstructions rely on both G. bulloides and G. ruber  jointly being used when calculating summer and winter SST's. This entails that we cannot calculate a summer SST just based on G. ruber.*

In fact, a multi-proxy comparison of SST reconstructions derived from different tracers is complex enough to discuss a bit more the uncertainties associated with each SST estimate (alkenones, ruber Mg/Ca-based and bulloides Mg/Ca-based) prior to work around with the equations you elaborate in Figure 2. My overall feeling is that the use of the equations to compute winter vs. summer SST is that, in the end, adds more confusion than resolve some uncertainties, as long as a clear comparison/evaluation between modern SST variability (that should be shown somehow in a figure) and the winter vs. summer SST calculation for core tops, which makes me doubtful that the equations used are really adding some values w.r.t. a more basic use of multi-species SST estimates.

*In relation to our seasonal temperature approach, please see response to earlier comment. With respect to uncertainties in our SST reconstructions, we are grateful for this comment. We have added a passage in the text and visualized error envelopes for all SST records in figure 4.*

Additional editor comments
…"I am puzzled by the 40ka warming, with temperatures above LIG and Holocene values!"

*We are grateful for these comments and have noted the 40ka warming as well. We are currently preparing a follow up manuscript that will take our analyses of the SST estimates from core NIOP 929 further, which will include an in-depth assessment of the above-mentioned warming event as well. We would therefore defer a discussion of this event to the envisaged paper.*

…"In addition, I do not agree with you that some of your records share similarities with EPICA d18O."

*Having now re-read the manuscript, we agree that the statement referring to the relation with AA records was a little to general. We have rephrased the respective section.*